

# Invasions of gladiolus rust in North America are caused by a widely-distributed clone of *Uromyces transversalis*

Jeffery A. DeLong[1], Jane E. Stewart[1,2], Alberto Valencia-Botín[3], Kerry F. Pedley[4], James W. Buck[5] and Marin T. Brewer[1]

[1] Department of Plant Pathology, University of Georgia, Athens, GA, United States of America
[2] Department of Bioagricultural Sciences and Pest Management, Colorado State University, Fort Collins, CO, United States of America
[3] Centro Universitario de la Ciénega, Universidad de Guadalajara, Ocotlán, Jalisco, Mexico
[4] Foreign Disease-Weed Science Research Unit, United States Department of Agriculture Agricultural Research Service (USDA-ARS), Fort Detrick, MD, United States of America
[5] Department of Plant Pathology, University of Georgia, Griffin, GA, United States of America

Corresponding author
Marin T. Brewer, mtbrewer@uga.edu

## ABSTRACT

*Uromyces transversalis*, the causal agent of Gladiolus rust, is an invasive plant pathogen in the United States and is regulated as a quarantine pathogen in Europe. The aim of this research was to: (i) determine the origin of introductions of *U. transversalis* to the United States, (ii) track the movement of genotypes, and (iii) understand the worldwide genetic diversity of the species. To develop molecular markers for genotyping, whole genome sequencing was performed on three isolates collected in the United States. Genomes were assembled *de novo* and searched for microsatellite regions. Primers were developed and tested on ten isolates from the United States resulting in the identification of 24 polymorphic markers. Among 92 isolates collected from Costa Rica, Mexico, New Zealand, Australia, and the United States there were polymorphisms within isolates with no genotypic diversity detected among isolates; however, missing data among the New Zealand and Australia isolates due to either poor amplification of degraded DNA or null alleles as a result of genetic differences made it difficult to generate conclusions about these populations. The microsatellite loci and flanking regions showed high diversity and two divergent genomes within dikaryotic individuals, yet no diversity among individuals, suggesting that the invasive *U. transversalis* populations from North America are strictly clonal.

## INTRODUCTION

Gladiolus rust, caused by the fungus *Uromyces transversalis,* was first identified in South Africa by von Thümen in 1876; however, little is known about the genetic diversity, center of origin, or historical dispersal patterns of *U. transversalis*. It was not until about a century after it was initially described that the fungus invaded northern Africa and then

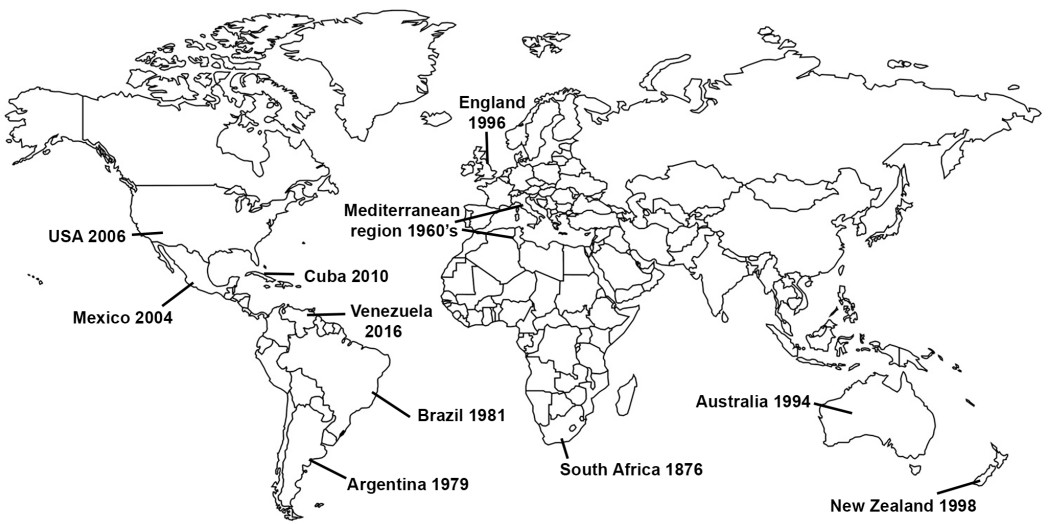

**Figure 1** Invasion history of the Gladiolus rust fungus, *Uromyces transversalis*.

southern Europe in 1966 (Fig. 1), reaching England by 1996 (*Beilharz, Parbery & Pascoe, 2001*). Subsequently, Gladiolus rust was detected in Argentina in 1979 (*Lindquist, Alippi & Medera, 1979*), Brazil in 1981 (*Pita et al., 1981*), Australia in 1994 (*Beilharz, Parbery & Pascoe, 2001*), New Zealand in 1998 (*McKenzie, 2000*), Mexico in 2004 (*Rodríguez-Alvarado et al., 2006*), the United States (USA) in 2006 (*Blomquist et al., 2007*), Cuba in 2010 (*Martínez-de la Parte et al., 2011*) and Venezuela in 2016 (*Mohali & Aime, 2018*). *Uromyces transversalis*, the causal agent of Gladiolus rust, can be devastating to species of *Gladiolus* and is difficult to eradicate once established.

The fungus *U. transversalis* is an obligate biotrophic pathogen that grows and reproduces on members of the family Iridaceae in arid, Mediterranean, and tropical climates (*Garibaldi & Aloj, 1980*; *Hernández, 2004*; *Peterson & Berner, 2009*; *Rizvi et al., 2007*). In regions where Gladiolus rust is established the disease can cause crop losses of 10–100%, unless fungicide applications are used (*Beilharz, Parbery & Pascoe, 2001*; *Ferreira & Nevill, 1989*; *Hernández, 2004*; *Littlejohn & Blomerus, 1997*; *Valencia-Botin et al., 2013*). As a consequence, the pathogen is considered of quarantine significance in Europe and was regulated in the USA from 2007 to 2015 (*Peterson & Berner, 2009*; *Rizvi et al., 2007*).

*Gladiolus* flowers are imported into the USA from multiple countries including Mexico, where *U. transversalis* is prevalent in *Gladiolus* production areas (*Valencia-Botin et al., 2013*). Shipments of *Gladiolus* flowers infected with *U. transversalis* arriving to the United States from Mexico have been repeatedly intercepted at border stations in California and Texas (*Brown, 2005*; *Hernández, 2004*; *Rizvi et al., 2007*; *Valencia-Botin et al., 2013*), and at a Florida border station with imports arriving from Mexico and Brazil (*Schubert et al., 2007*). A quarantine and national management plan strategy was followed by both federal and state quarantine officials in an attempt to contain and manage *U. transversalis* in the USA (*Rizvi et al., 2007*; *Valencia-Botin et al., 2013*). Despite quarantine measures, severe outbreaks of Gladiolus rust occurred in 2014 in the United States, leading the U.S.

Department of Agriculture, Animal and Plant Health Inspection Services (USDA-APHIS) to revise its response requirements in 2015 (*US Department of Agriculture APHIS, 2015*).

*Uromyces transversalis* primarily infects the leaves and stem of its host; however, under heavy inoculum pressure it can also infect the flowers (*Ferreira & Nevill, 1989*). Visibly infected plants lose economic value as an ornamental cut flower (*Valencia-Botin et al., 2013*). Infection by the rust fungus reduces the plant's vigor, resulting in reduced flower production (*Wise, Mueller & Buck, 2004*). The initial symptoms of *U. transversalis* on *Gladiolus* leaves are small chlorotic spots, which eventually break the leaf surface to reveal small yellow-orange uredinia. The uredinia coalesce to form large lesions (3–7 mm) laterally across the leaf surface (*Beilharz, Parbery & Pascoe, 2001*; *Martínez-de la Parte et al., 2011*; *Rizvi et al., 2007*; *Rodríguez-Alvarado et al., 2006*; *Valencia-Botin et al., 2013*). *U. transversalis* produces urediniospores and teliospores, but has no known alternate host (*Hernández, 2004*) or other spore types (*Hernández, 2004*; *Rizvi et al., 2007*). This suggests that sexual reproduction does not occur in *U. transversalis* due to an incomplete life cycle. As with many rusts, the urediniospores are the dispersal and infection spores. *U. transversalis* spores may be disseminated locally by wind or water splash (*Hernández, 2004*). Long-distance dispersal of urediniospores may occur naturally by wind, but it is primarily attributed to human-mediated movement of infected plants, including potted flowers, cut flowers and corms (*Beilharz, Parbery & Pascoe, 2001*; *Wise, Mueller & Buck, 2004*).

Molecular markers for genotyping isolates are necessary to understand the genetic diversity and historical dispersal patterns of *U. transversalis*. Due to the high variability, multiplexing capacity, ease of reproducibility, and relatively low cost associated with processing a large number of isolates (*Frenkel et al., 2012*; *Leclercq, Rivals & Jarne, 2007*), microsatellite markers are the ideal marker choice for determining the genetic diversity and population structure of *U. transversalis*. The objectives of this research were to: i) develop microsatellite markers to genotype isolates of *U. transversalis*, ii) determine the geographic origin and track the movement of introduced genotypes of *U. transversalis* in the USA, and iii) understand the genetic diversity of *U. transversalis* collected from a wide geographical area in order to understand historical dispersal patterns of this invasive fungus. We hypothesize that *U. transversalis* was introduced into the USA from Mexico, and that the invasive populations have low genetic diversity.

## MATERIALS & METHODS

### Isolate collection and DNA extraction

DNA was extracted from a total of 92 samples of *Uromyces transversalis* (Table 1) in preserved leaf tissue obtained from Australia ($n = 7$), New Zealand ($n = 10$), Mexico ($n = 60$), or as fresh, infected leaf tissue collected within the USA ($n = 10$) or from border interceptions from Costa Rica ($n = 5$). The preserved tissue from Australia and New Zealand had been stored in envelopes in herbaria, whereas the tissue from Mexico had been stored individually in conical polypropylene tubes at $-20\ °C$. From each preserved leaf tissue sample, DNA was extracted using a modified genomic DNA miniprep protocol (*Lee, Milgroom & Taylor, 1988*). Briefly, multiple uredinia were scraped to

**Table 1** Location and sources of *Uromyces transversalis* isolates used in this study.

| Geographic origin | | Original host species | Isolate identifier | Collection date | Culture Collection (Collector) |
|---|---|---|---|---|---|
| **City and/or State** | **Country** | | | | |
| Not provided | Costa Rica | *Gladiolus* sp. | CR497224, CR498594, CR498400, CR497666, CR498457 | 2012 | USDA-ARS; Pedley, K. |
| Wellington | New Zealand | *Gladiolus* sp. | NZ71109 | 2000 | New Zealand Fungal Herbarium (NZFB); Beever, R. |
| Fendalton | New Zealand | *Gladiolus* sp. | NZ87970 | 2006 | NZFB; Close, R. |
| Remuera | New Zealand | *Anomatheca laxa* | NZ69482 | 1998 | NZFB; Dingley, J.M. |
| Remuera | New Zealand | *G. nanus* | NZ69481 | 1998 | NZFB; Heckler, R. |
| Feilding | New Zealand | *Gladiolus* sp. | NZ71696 | 2000 | NZFB; Hill, C.F. |
| Mount Albert | New Zealand | *G. undulatus* | NZ97335 | 2007 | NZFB; Petley, M. |
| Avondale | New Zealand | *Melasphaerula* | NZ69208 | 1998 | NZFB; Wilkie, J.P. |
| Mount Albert | New Zealand | *Gladiolus* sp. | NZ99990 | 2011 | NZFB; Wilkie, J.P. |
| Mount Albert | New Zealand | *Tritonia* | NZ88195 | 2004 | NZFB; Wilkie, J.P. |
| Mount Albert | New Zealand | *Tritonia* | NZ69483 | 1998 | NZFB; Beever, R. |
| Not provided | Australia | *Gladiolus* sp. | VPRI 20841, VPRI 20858, VPRI 20881, VPRI 21344, VPRI 22299, VPRI 32661 | Unknown | Victoria Plant Pathology Herbarium (VPPH) |
| Mont Albert, Victoria | Australia | *Gladiolus* sp. | VPRI 21238 | 1996 | VPPH; Parbery, D. |
| Tlapizalco, Zumpahuacán | Mexico | *Gladiolus* sp. | TLAP1, TLAP2, TLAP3 | 2011 | Valencia-Botín, A. |
| Atlixco, Puebla | Mexico | *Gladiolus* sp. | Atlix1, Atlix2, Atlix3 | 2011 | Valencia-Botín, A. |
| Cuautla, Morelos | Mexico | *Gladiolus* sp. | Cua1, Cua2, Cua3 | 2011 | Valencia-Botín, A. |
| Villa Guerrero, State of Mexico | Mexico | *Gladiolus* sp. | Gro1, Gro2, Gro3 | 2011 | Valencia-Botín, A. |
| Tenango del Valle, State of Mexico | Mexico | *Gladiolus* sp. | Ten1, Ten2, Ten3 | 2011 | Valencia-Botín, A. |
| Irimbo, Michoacán | Mexico | *Gladiolus* sp. | Iri1, Iri2, Iri3 | 2010 | Valencia-Botín, A. |
| "La Finca" Villa Guerrero, State of Mexico | Mexico | *Gladiolus* sp. | LF1 1, LF1 2, LF1 3, LF2 1, LF2 2, LF2 3 | 2011 | Valencia-Botín, A. |
| Cocoyoc Yautepec, Morelos | Mexico | *Gladiolus* sp. | M1 1, M1 2, M1 3 | 2010 | Valencia-Botín, A. |
| Oacalco Yautepec, Morelos | Mexico | *Gladiolus* sp. | M2 1, M2 2, M2 3 | 2010 | Valencia-Botín, A. |
| Yautepec Yautepec, Morelos | Mexico | *Gladiolus* sp. | M3 R1, M3 R2, M3 R3 | 2010 | Valencia-Botín, A. |
| El Caracol Yautepec, Morelos | Mexico | *Gladiolus* sp. | M4 1, M4 2 | 2010 | Valencia-Botín, A. |
| Villa Ayala, Morelos | Mexico | *Gladiolus* sp. | M5 R3, M6 R1, M6 R2, M6 R3, M7 1, M7 2, M7 3, M8 R1, M8 R2, M8 R3 | 2010 | Valencia-Botín, A. |

**Table 1** (*continued*)

| Geographic origin | | Original host species | Isolate identifier | Collection date | Culture Collection (Collector) |
|---|---|---|---|---|---|
| **City and/or State** | **Country** | | | | |
| Ejido Tlayacapan, Morelos | Mexico | *Gladiolus* sp. | M9 R1, M9 R2, M9 R3 | 2010 | Valencia-Botín, A. |
| Huachinanitla Tepoztlán, Morelos | Mexico | *Gladiolus* sp. | M10 R1, M10 R2, M10 R3 | 2010 | Valencia-Botín, A. |
| Villa Guerrero, State of Mexico | Mexico | *Gladiolus* sp. | JB2, JB7 | 2010 | Valencia-Botín, A. |
| Cuautla, Morelos | Mexico | *Gladiolus* sp. | JB3, JB5 | 2010 | Valencia-Botín, A. |
| Atlixco, Puebla | Mexico | *Gladiolus* sp. | JB1, JB4, JB6 | 2010 | Valencia-Botin, A. |
| Irambo, Michoacan | Mexico | *Gladiolus* sp. | JB8 | 2010 | Valencia-Botin, A. |
| California | United States | *Gladiolus* sp. | CA11-1 | 2011 | K. Pedley |
| Carpenteria, California | United States | *Gladiolus* sp. | CA14-1, CA14-3, CA14-4 | 2014 | K. Pedley |
| Santa Maria, California | United States | *Gladiolus* sp. | CA14-2 | 2014 | K. Pedley |
| Santa Barbara, California | United States | *Gladiolus* sp. | CA14-5 | 2014 | K. Pedley |
| Goleta, California | United States | *Gladiolus* sp. | CA14-6, CA14-7 | 2014 | K. Pedley |
| Manatee County, Florida | United States | *Gladiolus* sp. | FL11-1 | 2011 | K. Pedley |
| Hendry County, Florida | United States | *Gladiolus* sp. | FL11-2 | 2011 | K. Pedley |

remove urediniospores (0.02–0.04 g) using a sterilized scalpel and transferred into 1.5 mL reaction tubes containing 246.9 µL lysis buffer (150 µL sddH$_2$O, 25 µL 0.5 M EDTA (pH 8), 25 µL 1.0 M Tris, 43.75 µL 20% SDS solution, 3.15 µL 20 mg/L proteinase K and 0.0025 g NaHSO$_3$) (Thermo Fisher Scientific, Watham, MA). Sample tubes were vortexed for 1 min, incubated at 65 °C for 15 min, and centrifuged (13,978× g for 5 min). The precipitates were discarded and the supernatants transferred to new 1.5 µL microcentrifuge tubes. A solution of 50 µL 7.5 M NH$_4$OAc (Sigma-Aldrich, St. Louis, MO) was added to each tube, vortexed for 10 s, and tubes were chilled on ice for 15 min. Samples were then centrifuged (13,978× g for 3 min). The supernatants were again transferred to new 1.5 mL microcentrifuge tubes and 175 µL isopropanol was added, tubes were mixed and centrifuged (13,978× g for 5 min), and the supernatant was discarded. The pellets were rinsed twice with 250 µL of 70% ethanol solution, dried and re-suspended with 25 µL sddH$_2$O, then incubated at 30 °C for 10 min. All samples were stored at −20 °C until further use. DNA extractions from fresh, infected leaf tissue was performed using a modified hexadecyltrimethylammonium bromide (CTAB) protocol (*Rodgers & Bendich, 1985*). Briefly, one to three one cm$^2$ excised pieces of infected leaf tissue were frozen and using a mortar and pestle, ground to a fine powder in liquid nitrogen. DNA extraction buffer (100 mM Tris, pH 7.5; 1% CTAB; 0.7 M NaCl; 10 mM EDTA; 1% 2-mercaptoethanol; 0.3 mg/ml proteinase K) was added to the ground tissue and incubated at 65 °C for 30 min, followed by two rounds of chloroform:

isoamyl alcohol (1:1) extraction, and precipitated with 2-propanol. DNA was resuspended in TE buffer containing 1 mg/ml RNase.

Three *U. transversalis* isolates collected in the USA were maintained at the USDA Agricultural Research Service, Foreign Disease-Weed Science Research Unit, biosafety level-3 plant disease containment facility at Ft. Detrick, Maryland. These samples were propagated from *U. transversalis-* infected Gladiolus plants collected from California and Florida commercial fields in 2011 and 2014 (Table 1). Prior to extraction using the modified CTAB protocol described above, urediniospores were harvested from infected Gladiolus plants with a microcyclone spore collector (*Cherry & Peet, 1966*; *Peterson & Berner, 2009*; *Tervet et al., 1951*). Spores were germinated by placing 300 mg of freshly harvested spores in a 23 cm × 33 cm glass container that contained 300 mL of sterile water with 15 µg ampicillin. A sterile wooden applicator stick was used to break up clumps of spores, so that the spores were evenly distributed across the surface of the water. The container was covered and left in the dark overnight (16–18 h). The germinated spores were then scraped from the surface of the water, blotted dry with sterile-paper towels and stored in −20 °C.

## Genome sequencing and assembly

Genomic DNA obtained from three isolates (CA11-1, FL11-1, and FL11-2) was standardized to 50.0 ng/µL using a nanodrop and sent to the Georgia Genomics Facility (GGF) (University of Georgia, Athens, GA) for library preparation and sequencing using the Illumina MiSeq platform as 300-bp paired ends reads using a 600 cycle cartridge with a NGS library preparation method. The raw forward and reverse reads of each isolate was observed using FASTQC v.11.2 (Babraham Bioinformatics Institute). Quality control was performed using FASTX-Toolkit v.3.0.13 (http://hannonlab.cshl.edu/fastx_toolkit/). All reads with a phred score below $Q = 22$ were removed prior to assembly. ABySS v.1.3.6 (*Simpson et al., 2009*) was used for *de novo* assembly of forward and reverse reads into contiguous sequences (contigs) for each isolate, using an optimal K-mer value of 64 determined with multiple assembly trials. Generated contigs were then imported into Geneious v.6.1.8 (*Kearse et al., 2012*).

## Microsatellite discovery and marker development

To increase the potential for successful microsatellite marker development, only contigs 200 bp or greater in size with matched pair reads were considered. Contigs and singletons were searched for at least five perfect repeats of trimeric, tetrameric, pentameric, and hexameric motifs in Geneious v.6.1.8 using Phobos v.3.3.12 (*Kearse et al., 2012*; *Mayer, 2006*). Mono and dinucleotide repeats were eliminated due to the difficulty of scoring allele differences. Contigs with microsatellites identified using our criteria were aligned by multiple sequence alignment using Geneious Align v.6.1.8. default parameters. Microsatellites shared among the three isolates were visually assessed for sequence variation. Those that showed microsatellite repeat number variation among isolates and had at least 50 bp flanking each side of the repeat were considered acceptable for primer design. Primers for amplification of microsatellite loci were designed with Primer3web v.4.0 (*Koressaar & Remm, 2007*; *Rozen & Skaletsky, 1999*; *Untergasser et al., 2012*) to produce amplicons of approximately 180–350 bp in length with an optimal annealing temperature of 59 °C.

Sixty primer pairs for candidate microsatellite loci were initially evaluated on the three sequenced isolates CA11-1, FL11-1, and FL11-2 to verify that the PCR worked with the designed primers and that the PCR amplicons were the expected size. PCR was carried out in 10 μL reactions with 1 μL of 10 × ExTaq buffer (Takara Bio Inc., Mountain View, CA), 1 μL of 2.5 mM dNTPs (Takara Bio Inc.), 0.25 μL of 10 μM forward primer, 0.25 μL of 10 μM reverse primer (Integrated DNA Technologies, Coralville, IA), 0.1 μL of ExTaq polymerase (Takara Bio Inc.), 6.9 μL of sterile distilled $H_2O$, and 0.5 μL of 50.0 ng/μL DNA template. Reaction conditions were 94 °C for 2 min followed by 35 cycles of denaturation at 94 °C for 30 s, annealing at 59 °C for 30 s and extension at 72 °C for 30 s, followed by a final extension of 72 °C for 5 min. Amplification of PCR products within the expected size range was confirmed by electrophoresis run at 95 V (4.75 V/cm) on a 2% (wt/vol) agarose gel (Alfa Aesar, Haver Hill, MA) for 2.5 h using a 100 bp size standard (New England Biolabs Inc., Ipswich, MA).

Twenty-five primer sets (Table 2) that successfully amplified the three sequenced isolates were screened for polymorphism on a panel of *U. transversalis* that included seven additional isolates from California (Table 1). A three-primer method (*Schuelke, 2000*) was used in this round of marker evaluation. The forward primer for each candidate marker had a CAG tag (5′-CAGTCGGGCGTCATCA-3′) (*Hauswaldt & Glenn, 2003*) added to the 5′ end. The third primer consisted of the CAG tag, labeled with a 6FAM fluorescent dye (Invitrogen Inc., Carlsbad, CA) on the 5′ end. PCR was carried out in 12 μL reactions with 1.2 μL of 10 × ExTaq buffer (Takara Bio Inc., Mountain View, CA), 1.2 μL of 2.5 mM dNTPs (Takara Bio Inc.), 0.1 μL of 10 μM forward primer, 0.5 μL of 10 μM reverse primer (Integrated DNA Technologies, Coralville, IA), 0.5 μL of 10 μM 5′6FAM-labeled CAG tag primer (Invitrogen Inc.), 0.1 μL of ExTaq polymerase (Takara Bio Inc.), 7.9 μL of sterile distilled $H_2O$, and 0.5 μL of approximately 50.0 ng/μL DNA template. Reaction conditions were 94 °C for 2 min followed by 35 cycles of denaturation at 94 °C for 30 s, annealing at 55 °C for 30 s and extension at 72 °C for 30 s, followed by a final extension of 72 °C for 5 min. Amplification of individual PCR products within the expected size range was confirmed by electrophoresis.

One microliter of a 1:10 dilution of PCR product was added to 0.1 μL of GeneScan 500 LIZ-labeled size standard and 9.9 μL of Hi-Di formamide (Applied Biosystems Inc., Foster City, CA). Amplicons were denatured by incubation at 95 °C for 5 min and immediately placed on ice. Fragment analysis was conducted at the GGF on an Applied Biosystems 3730xl 96-capillary DNA Analyzer. GeneMapper v.4.0 (Applied Biosystems Inc.) and Geneious v.6.1.8 (*Kearse et al., 2012*) were used to determine allele sizes from the chromatograms.

## Multiplex PCR

Eight primer sets (Table 2, see loci with asterisks) that consistently produced fragments within the expected size range for the 10 isolates were optimized for multiplex PCR. Two multiplex reactions (multiplex 1 –*Ut513*, *UtCA759*, *Ut2648* and *Ut3161*; multiplex 2 –*Ut497*, *Ut1841*, *Ut1908* and *Ut2048*) were developed to increase efficiency and decrease cost for genotyping a large panel of isolates. The forward primers of the microsatellite markers selected for multiplex PCR were labeled at the 5′ end with one of the fluorescent

**Table 2  Repeat motif, primer sequences, and number of alleles, allele sizes, and genotypes for 25 *U. transversalis* microsatellite markers.**

| Locus[a] | Repeat motif | Primer sequence (5′→3′)[b] | Number of observed alleles, allele sizes (bp), and genotypes[c] |
|---|---|---|---|
| *Ut337* | $(AGG)_7$ | F: CGGAAGAGATGAGTGGTCAAG | 2 (195, 198) |
| | | R: TCACATCATCCCCTCCCTA | |
| *Ut397* | $(TTG)_9$ | F: TTCGATTCGATTCGTTTGTTT | 1 (259) |
| | | R: GGATGTTTTGATTCTGTTAGAGAGTG | |
| *Ut447* | $(ACC)_6$ | F: TGCTTCAGCTTCCCAAAACT | 2 (237, 240) |
| | | R: TGGCTGTGAATTGTGAGACC | |
| *Ut497*2* | $(GAA)_{15}$ | F: CTTGAAGGGGATCGAGAAGA (6FAM) | 2 (232, 251) |
| | | R: TGTTCTCCGGCAGAGGTTTA | |
| *Ut513*1* | $(TCA)_6$ | F: TCCCAAACAAATCGTGAAGA (NED) | 2 (200, 203) |
| | | R: GCTCCCGTTAATGGTCACAG | |
| *Ut542* | $(GTT)_5$ | F: GTCTTCTTTGCTGCGTTTCC | 2 (204, 207) |
| | | R: TCCTGGTTTTGAACCTCCTG | |
| *Ut568* | $(ACC)_6$ | F: TCCCATGGGTTTGGTTGC | 2 (178, 181) |
| | | R: TCCTTAATCTGGGTTGACATTT | |
| *Ut575* | $(TTA)_5$ | F: TGACGATCCTAACGAAGGGTA | 2 (241, 244) |
| | | R: CTTGGGGTACGAGAGCACTT | |
| *Ut697* | $(AAG)_5$ | F: TAGGCGAAGTGGTACGAGGT | 1 (224) |
| | | R: AGGGAAGAAGAGGGTCAACA | |
| *Ut752* | $(ATC)_6$ | F: AGTCTTGTGCTGGTCTTCGTC | 2 (213, 216) |
| | | R: TTTGCCGCCTTATATTGTCA | |
| *Ut844* | $(ACT)_8$ | F: CTCCGTCAGCCAGTCAGTC | 1 (310) |
| | | R: GATGAGGTTGAGGGCGAGTA | |
| *Ut981* | $(TGA)_6$ | F: GGGTCAAACAGGTCTTCTGG | 1 (202) |
| | | R: CTACTGAAATGGGCCACAAA | |
| *Ut1272* | $(AAG)_5$ | F: TGAAGTTTTCCACCCTGGTT | 2 (253, 256) |
| | | R: ATCTTGGGCAAACTGACCAC | |
| *Ut1289* | $(GAG)_7$ | F: GGTCTTGAGAGAACGGAGGA | 2 (254, 257) |
| | | R: CTCTTCCAGATACCCCACCA | |
| *Ut1841*2* | $(AGG)_5$ | F: GAACCCTGCCTCACACCTTA (NED) | 2 (345, 348) |
| | | R: GCGGCTACCAGAGCTTTAGA | |
| *Ut1908*2* | $(GAT)_6$ | F: TCCTCTCAGCCAATCCAATC (PET) | 2 (200, 203) |
| | | R: CTCTTGCCCATCAATCCAAC | |
| *Ut2035* | $(TTTA)_8$ | F: GGATCGAGTCGGTCGATTTA | 2 (229, 232) |
| | | R: GCCGAACAGGACTAGCATTG | |
| *Ut2048*2* | $(GAA)_6$ | F: CGAGCGATAAATTTTTGAACA (VIC) | 2 (182, 185) |
| | | R: TGTCCGGAGAATGTGAACTG | |
| *Ut2443* | $(GAA)_8$ | F: AGAATTGGATGAAACAGGGAGA | 1 (188) |
| | | R: AAGGAGGAAGCCATCACTCA | |
| *Ut2536* | $(GAG)_5$ | F: AGGGCTGGTAGACGTGACTG | 2 (248, 251) |
| | | R: TCATGTCTCTGACACCACCA | |

**Table 2** (*continued*)

| Locus[a] | Repeat motif | Primer sequence (5′→3′)[b] | Number of observed alleles, allele sizes (bp), and genotypes[c] |
|---|---|---|---|
| *Ut2648*[*1] | $(CAG)_6$ | F: GAACTGGTGCAACCGATACA (VIC) | 2 (266, 269) |
| | | R: CACAGCCTTGGCTCTTGAGT | |
| *Ut3161*[*1] | $(TCC)_6$ | F: GAGTCTGGCCCAGCTGTTT (6FAM) | 2 (192, 195) |
| | | R: TCTGATCTTGCAGGGGATTC | |
| *UtCA759*[*1] | $(CAT)_7$ | F: GATGGCCAGAAGAAAGATGC (PET) | 1 (296) |
| | | R: TTAACCAGCGCGAGAGTCTT | |
| *UtCA809* | $(TTA)_7$ | F: GCCACTTCTCCAAACGCTTA | 1 (258) |
| | | R: TCGCAAGATCAAGAAACAACC | |
| *UtCA950* | $(GTT)_9$ | F: GGCAGAGGATGAGTCGTGTA | 2 (272, 287) |
| | | R: TCATCTCATCCCCACAATCA | |

**Notes.**

[a]Asterisks indicate loci that were used for the multiplex reactions and the 1 or 2 indicate multiplex 1 or 2, respectively.

[b]The fluorescent dye used for multiplex reactions is listed in parentheses to the right of the forward primer.

[c]Genotype of all 10 isolates from the United States. Allele sizes are listed based on the results of the multiplex reactions or what the length of the alleles would be without the 16 nucleotide CAGTAG.

dyes from the DS-33 dye set: 6-FAM (Integrated DNA Technologies), VIC, PET, or NED (Applied Biosystems Inc.). Multiplex reactions were optimized so that loci with alleles of similar size ranges were labeled with different dyes. All 92 samples were genotyped with the eight markers in the multiplex reactions.

Multiplex PCR was conducted using a modified protocol of the Type-it Microsatellite PCR kit (Qiagen, Hilden, Germany) in 10 μL reactions with 5 μL of 2 × Type-it Master Mix buffer, 1 μL of 10 × primer mix (2 μM of each primer in the multiplex), 3 μL of sterile distilled $H_2O$, and 1 μL of approximately 50.0 ng/μL DNA template. Reaction conditions were 94 °C for 2 min followed by 35 cycles of denaturation at 94 °C for 30 s, annealing at 55 °C for 30 s and extension at 72 °C for 30 s, followed by a final extension of 72 °C for 5 min. Amplification of PCR products within the expected size range was confirmed by electrophoresis. The PCR products were prepared as described above and sent to GGF for fragment analysis. Geneious v.6.1.8 (*Kearse et al., 2012*) was used to determine allele sizes from the chromatograms. Loci were distinguished by fluorescent dye. Only peaks above the relative intensity cutoff threshold of 500 relative fluorescence units (RFU) were scored.

## RESULTS

### Whole genome sequencing and assembly

The Illumina MiSeq PE 300 sequencing platform generated 32,461,282 reads with an average insert size of 575 bp and read lengths of 301 bp. Sequence quality was assessed by phred score and signal purity filter values resulting in a total of 25,452,493 reads with a P F of 99.26%, which corresponds to 6.02 to 9.98 million reads for CA11-1, FL11-1, and FL11-2 (Table 3).

The *de novo* draft assemblies resulted in 5,706,372 (94.7% assembled out of the total filtered reads), 4,305,978 (43.2%), and 7,444,849 (80.4%) total assembled reads for CA11, FL11-1, and FL11-2, respectively. This Whole Genome Shotgun project has been deposited at DDBJ/ENA/GenBank under the accessions PTJR00000000, PTJQ00000000,

**Table 3  Genome assembly and microsatellite statistics.**

| Isolates | Total reads | Reads after quality filtering[a] | Reads that assembled | # Contigs > 200 bp | Contigs w/ microsatellites[b] | % Microsatellites per assembly |
|---|---|---|---|---|---|---|
| CA11-1 | 7,762,942 | 6,023,634 | 5,706,372 | 466,181 | 4,599 | 0.98% |
| FL11-1 | 12,804,893 | 9,976,981 | 4,305,978 | 548,017 | 5,685 | 1.03% |
| FL11-2 | 11,893,445 | 9,262,312 | 7,444,849 | 645,533 | 8,666 | 1.34% |

Notes.

[a]Based upon purity filter value of 99.26%.

[b]Contigs with identified microsatellites based on the annotation criteria: repeat unit length = min: 3 max: 6, min. length of 15. Mono and dinucleotide repeats not considered due to the difficulty of scoring alleles during fragment analysis.

and PTJP00000000 for CA11, FL11-1, and FL11-2, respectively. Using Geneious v.6.1.8 (*Kearse et al., 2012*), contigs for each isolate were filtered to select only those 200 bp in length or greater. This resulted in 466,181, 548,017, and 645,533 contigs for CA11, FL11-1, and FL11-2, respectively, which were subsequently used to search for microsatellite repeats (Table 3).

## Microsatellite marker development

An alignment of the 18,950 contigs produced 4,296 contigs with potentially informative microsatellite loci shared among the three isolates. Microsatellite loci shared by at least two of the three isolates were observed in 2,754 of the aligned contigs. Microsatellites were identified in 0.98%, 1.03%, and 1.34% of the contigs, showing that the discovery rate of microsatellites was consistent among sequenced isolates.

Sixty sets of primers were developed and screened by PCR on the three isolates of *U. transversalis*. Of the 60 putative markers, 25 were successfully amplified by PCR and evaluated for polymorphism on the panel of ten isolates from the United States (Tables 1, 2). To determine if contigs used for marker development were sequences of *U. transversalis* and not contaminant sequence, we used blastn at NCBI (*Altschul et al., 1990*) to identify the sequences in the database that were most similar with high coverage. Most often the top hits were *Puccinia graminis*, another rust species, or another fungus, with percent identity always <90%. Sometimes the top hit was another eukaryote, but it was an unlikely contaminant, such as mouse, sheep or fish. The percent identity for these was always <90% and the e-values were usually high (>0.01) or the query coverage was low (<10%). We concluded that the contigs selected for marker development were sequences of *U. transversalis* and not contaminants. Isolate CA14-7 consistently produced peaks below our cutoff of 500 RFU for reliable data; however, there were fragments at the expected allele size ranges. It is possible that there were PCR inhibitors in the DNA extract or that the DNA was of lower quality than the other samples. Of the 25 markers, seven were monomorphic, with only one allele each, and 18 were polymorphic, with two alleles each (Table 2). All 10 isolates from the United States were the same genotype based on the 25 markers. The polymorphism was identified among alleles within each locus, rather than among individuals. Overall, the microsatellite markers showed allelic diversity, but no genotypic diversity among the isolates from the United States. There was a high heterozygosity within individuals with each isolate having both alleles for the polymorphic loci (Table 2).

## Genotypic analyses

Eight primer sets that consistently produced fragments within the expected size range for the 10 isolates were optimized for multiplex PCR. We selected markers that were polymorphic and did not produce alleles in overlapping size ranges. Unfortunately, after subsequent genotyping runs one of the markers *UtCA759* was determined to be monomorphic. Even though the four markers within each multiplex were run with different fluorescent dyes, we minimized size overlap to prevent pull-up effects or bleed through from the fragments run with the other dyes. When using the eight microsatellite markers in two multiplex reactions, samples from the USA, Costa Rica and Mexico consistently produced PCR products of the expected size. Samples from New Zealand and Australia produced inconsistently sized PCR products despite duplicate reactions. Using the same eight markers, we attempted to genotype 16 leaf samples from South Africa, but these repeatedly failed to produce PCR products. Only one isolate, PREM 57128, which was sampled in 1998 produced a faint PCR product; however, no fragments were detected in the analysis.

Fragment analysis showed allelic variation within individuals, but no genotypic variation was observed among the isolates from Australia, Costa Rica, Mexico, New Zealand, and the USA. In all cases where fragments were observed and were above the relative intensity cutoff of 500 RFU, the genotypes were identical to each other and to all isolates from the USA (Table 2). In some cases where the peaks were below the relative intensity cutoff of 500 RFU, there was a fragment for only one of the alleles or the alleles were a slightly different size; however, these results were not reliable (Table S1). Marker *Ut497* consistently failed to produce fragments or peaks above the relative intensity cutoff for almost all isolates where DNA was obtained from preserved leaf tissue. The six remaining markers (*Ut513*, *Ut1841*, *Ut1908*, *Ut2048*, *Ut2648* and *Ut3161*) were polymorphic with only two allele sizes observed for each marker, while one marker (*UtCA759*) was monomorphic, producing only one allele size.

## Sequence divergence

Visual assessment of the aligned contigs for the three sequenced isolates revealed that there were two distinct haplotypes (nucleotide sequence patterns) for each isolate occurring at nearly all microsatellite loci (Table 4, Fig. 2). The variation in repeat number occurred between alleles or haplotypes of the same isolate. Additionally, there were numerous single nucleotide polymorphisms (SNPs) detected in the regions flanking the microsatellite repeats. The two haplotypes within each of the three isolates sequenced were identical to the haplotypes among all three isolates for the loci compared, including microsatellite loci that were monomorphic based on sequence length. Variation in the microsatellite-flanking sequences between haplotypes was estimated for loci with complete data sets for the three isolates. The two alleles differed in nucleotide sequence by 1.6% to 6.9% (Table 4).

## DISCUSSION

There was no genotypic diversity observed among the *U. transversalis* isolates from Australia, Costa Rica, New Zealand, Mexico, and the USA based on the eight microsatellite loci developed in the present study. However, there was missing data for most of the

**Table 4  Variation between alleles within sequenced genomes of *U. transversalis*.**

| Microsatellite | No. Single nucleotide polymorphisms | No. Nucleotides in flanking regions | % Difference |
| --- | --- | --- | --- |
| *Ut337* | 3 | 180 | 1.7 |
| *Ut513* | 7 | 291 | 2.4 |
| *Ut568* | 13 | 187 | 6.9 |
| *Ut575* | 3 | 187 | 1.6 |
| *Ut752* | 9 | 263 | 3.4 |
| *Ut1908* | 10 | 264 | 3.8 |

**Figure 2  Comparison of partial sequences of the locus *Ut789* showing two distinct alleles for each of the three sequenced isolates.** While the image represents a single locus, a similar pattern was observed for the sequences of most microsatellite flanking regions. The sequences shown correspond with GenBank accession numbers PTJR01079144.1, PTJQ01464835.1, and PTJP01109568.1 for CA11-1, FL11-1, FL11-2, respectively, for the top genomes, and accession numbers PTJR01079145.1, PTJQ01463031.1, and PTJP01109569.1, for CA11-1, FL11-1, FL11-2, respectively, for the bottom genomes of the dikaryon. The coordinates are based on accession number PTJR01079144.1.

isolates from New Zealand and Australia, which could either be due to poor amplification of the low-quality DNA obtained from these dried herbarium samples or null alleles from poor primer binding as a result of genetic differences in these populations that could be biologically meaningful. For the ten isolates from the USA genotyped with all 25 microsatellite loci, eight of the markers were monomorphic and 16 were polymorphic, with all polymorphism observed within each of the isolates. Additionally, we genotyped some of the other isolates (five isolates from Costa Rica, one isolate from New Zealand, and three isolates from Mexico) at all 25 loci (Table 2) and still found no genotypic diversity. Since no genetic differences were observed among isolates with complete or nearly complete data, it is not possible to track individuals or introductions of individuals from other populations that may have occurred (*McDonald & Linde, 2002*; *Milgroom & Peever, 2003*). It may be possible to detect genotypic diversity within *U. transversalis* using different genotyping methods, such as whole genome sequencing (WGS) or genotyping-by-sequencing (GBS) for SNP detection; however, the large genome size of rust fungi or high heterozygosity present within individuals may obscure the detection of genotypic diversity among individuals (*Elshire et al., 2011*). Additionally, levels of genetic diversity and patterns of population genetic structure detected with microsatellite markers and SNPs derived from GBS have been found to be in agreement, although greater resolution of genotypes is usually possible

with the increased number of SNP loci generated by WGS or GBS, or similar methods such as RAD-seq (*Rafiei et al., 2018*; *Maurice et al., 2019*). A different member of the Pucciniales, *Phakopsora pachyrhizi*, for which only clonal reproduction has been observed, has some genotypic diversity both in the USA where it has been introduced and in Asia, where it is native (*Zhang et al., 2012*). Nonetheless, the microsatellite markers developed for *U. transversalis* may be useful for diagnostic purposes or detection of *U. transversalis* in asymptomatic plant material.

In this study, all of the genetic diversity identified in *U. transversalis* occurred as allelic diversity within individuals. Most of the microsatellite markers and all of the repeat-flanking sequence that were compared showed two alleles or two distinct haplotypes at each locus, indicative of very high heterozygosity. It is not entirely surprising that loci were polymorphic, since this fungus is a dikaryon, but it was surprising that all genotyped individuals shared the same genotype with fixed differences. Additionally, one would expect some of the loci to be homozygous following Hardy-Weinberg equilibrium in some individuals if the population were undergoing sexual reproduction. This level of allelic variation within each individual is suggestive of divergent genomes between the nuclei of the dikaryon across the invasive population. The flanking sequences from each locus showed approximately 97% sequence similarity (Table 4). In some cases, the two genomes of the dikaryon are more divergent than what is usually observed within a single fungal species (*Hibbett, 2016*). The lack of genotypic diversity among isolates and the distinct sequences and microsatellite alleles within individuals suggests that *U. transversalis* samples from Costa Rica, Mexico, and the USA are asexually reproducing populations that are not recombining through sexual reproduction (*Milgroom, 1996*). Clonal invasions are common among plants pathogens (*Milgroom et al., 2009*; *Goss, Carbone & Grünwald, 2009*). In dikaryotic organisms and diploids, the absence of sexual reproduction will increase the divergence between sequences in each genome as random mutations will occur over time (*Birky, 1996*). Thus, low genotypic diversity combined with high allelic diversity within individuals is suggestive of strict clonal reproduction for an extensive period of time (*Balloux, Lehmann & De Meeûs, 2003*; *Birky, 1996*). Our results provide support for clonal reproduction of *U. transversalis* in the USA, Mexico, and Costa Rica, which is consistent with the observed research on reproductive biology of *U. transversalis*.

To our knowledge this is the first study on the genetic diversity of *U. transversalis*. Since *U. transversalis* urediniospores are dikaryotic, development of codominant, sequence-specific microsatellite markers would be appropriate in addressing our questions of genetic diversity, origin, and sources of introductions for this rust fungus. Traditionally, microsatellite development required the construction of a genomic library enriched for repeated motifs, isolation, and sequencing clones; primer design and optimization; and testing for polymorphism on a few unrelated individuals (*Abdelkrim et al., 2009*; *Frenkel et al., 2012*; *Santana et al., 2009*; *Zhang et al., 2012*). As an alternate approach, the microsatellite markers in the present study were developed using whole genome sequencing of multiple isolates. This approach not only increased our chances of identifying polymorphic alleles within *U. transversalis*, but also supplied genomic sequence, which could be used for comparative analyses or other purposes. However, the main focus of this study was not

the whole genome sequencing and assembly, but to use these data to develop markers. Although useful for marker development, the three draft genomes were highly fragmented (4.3–7.4 million assembled reads or contigs compared to 6.0–9.9 million unassembled), which could be the result of repetitive DNA and a large genome, which are common among rust fungi (*Ramos et al., 2015*; *Tavares et al., 2014*).

Although we were unable to detect genetic diversity among isolates of *U. transversalis* across a wide geographic range, the genome sequences will serve as a resource for further studies on this destructive fungal pathogen of Gladiolus. Since all isolates sampled exhibited limited diversity and were genetically similar, this demonstrates that disease management strategies both current and future, should work for all locations and current hosts for *Uromyces transversalis*. However, there could be undetected variation in regions of the genome associated with virulence, host specificity, or fungicide resistance that are not linked to the microsatellite loci studied here. Future studies will help us to determine the usefulness of the microsatellite markers in diagnosis and detection.

## CONCLUSIONS

There was no genotypic diversity observed among the invasive *U. transversalis* populations from Australia, Costa Rica, New Zealand, Mexico, and the USA based on the eight microsatellite loci developed in the present study. However, there was missing data for most of the isolates from New Zealand and Australia, which could either be due to poor amplification of the low-quality DNA obtained from these dried herbarium samples or null alleles from poor primer binding as a result of genetic differences in these populations that could be biologically meaningful. The lack of genotypic diversity among isolates from North America and the distinct sequences and microsatellite alleles within individuals suggests that *U. transversalis* samples in introduced ranges are asexually reproducing populations that are not recombining through sexual reproduction. Our results provide support for clonal reproduction of *U. transversalis* in the USA, Mexico, and Costa Rica, which is consistent with the observed research on reproductive biology of *U. transversalis* and other invasive plant pathogens.

## ACKNOWLEDGEMENTS

We thank A. Ruck for her technical assistance in propagating isolates and preparing DNA extracts. Mention of trade names or commercial products in this publication is solely for the purpose of providing specific information and does not imply recommendation or endorsement by the US Department of Agriculture. USDA is an equal opportunity provider and employer.

### Funding

This work was funded by the University of Georgia and the United States Department of Agriculture, Agricultural Research Services (Non-Assistance Cooperative Agreement,

Project Number: 8044-22000-041-13-S). The funders had no role in study design, data collection and analysis, decision to publish, or preparation of the manuscript.

### Grant Disclosures
The following grant information was disclosed by the authors:
University of Georgia.
United States Department of Agriculture, Agricultural Research Services (Non-Assistance Cooperative Agreement, Project Number: 8044-22000-041-13-S).

### Competing Interests
The authors declare there are no competing interests.

### Author Contributions
- Jeffery A. DeLong conceived and designed the experiments, performed the experiments, analyzed the data, prepared figures and/or tables, authored or reviewed drafts of the paper, approved the final draft.
- Jane E. Stewart conceived and designed the experiments, analyzed the data, authored or reviewed drafts of the paper, approved the final draft.
- Alberto Valencia-Botín performed the experiments, contributed reagents/materials/-analysis tools, prepared figures and/or tables, approved the final draft.
- Kerry F. Pedley conceived and designed the experiments, performed the experiments, analyzed the data, contributed reagents/materials/analysis tools, authored or reviewed drafts of the paper, approved the final draft.
- James W. Buck and Marin T. Brewer conceived and designed the experiments, analyzed the data, contributed reagents/materials/analysis tools, prepared figures and/or tables, authored or reviewed drafts of the paper, approved the final draft.

### DNA Deposition
The following information was supplied regarding the deposition of DNA sequences:
This Whole Genome Shotgun project is available at DDBJ/ENA/GenBank: PTJR00000000, PTJQ00000000, and PTJP00000000 for CA11, FL11-1, and FL11-2.

### Data Availability
The raw genotype data is available in Table S1.

### Supplemental Information
Supplemental information for this article can be found online at http://dx.doi.org/10.7717/peerj.7986#supplemental-information.

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
