# Peer review of "Invasions of gladiolus rust in North America are caused by a widely-distributed clone of Uromycestransversalis"

_PeerJ, doi:10.7717/peerj.7986_

## Round 0.1 · original submission · Minor Revisions

Two reviewers provided detailed reviews of your manuscript. I agree with the reviewers that the supplemental documents illustrate key results and should be in the main manuscript. Please consider Reviewer 1's comments regarding what the failed markers indicate in terms of the diversity/clonality of this pathogen within and among regions and address them in your revision, and Reviewer 2's comments in the annotated pdf regarding two statements in the discussion that should be better supported. There are multiple other suggestions from the reviewers that address clarity and reporting. Addressing these will be of benefit to your readers.

Reviewer 1 ·

Basic reporting

No comment

Experimental design

No comment

Validity of the findings

No comment

Additional comments

The authors developed SSR markers to look at allelic and genotypic diversity in 92 isolates of Gladiolus rust. Genome sequence from three US isolates was used to design SSR markers. Markers were first tested on the three US isolates before checking them on seven additional isolates from US. Eight markers that performed well in the initial PCR screen were then multiplexed in two sets of four and run on all 92 isolates. The manuscript is well written and presented logically. However, I have few concerns that I list below.

Major concerns
----- --------
1. This is an obligate biotroph. How was it determined that the assembled regions belong to the rust fungus? I understand that the authors detailed in the methods section how the spores were collected and allowed to germinate on an antibiotic solution. But it would add to the manuscript if the authors check that the contigs selected for marker development are of fungal origin. There are several tools available to sort bacterial and fungal sequences or a simple BLAST might work.

2. How do you explain the negative results in Table S1 for Aus and NZ? It looks like that the negative results were ignored.
Not all Aus/NZ isolates group with US/MX/CR isolates. Three (Ut1841, Ut2048, Ut497) out of four markers in the second multiplex did not perform well. Even if for one moment the results from multiplex 2 are excluded and only look at results from markers in multiplex 1, only 3 and 4 isolates from NZ and AUS, respectively, have results from all four markers. What could be the reason?
Sample size? There are 5 isolates from Costa Rica but these all have similar genotypes.
Age of the sample? The NZ isolate NZ71696 was sampled in year 2000 performed better than other NZ samples collected in 2004 or 2006.
So, then how do you explain the lack of results for Aus and NZ, for markers that work well for isolates from US/MX/CR?

3. If the lack of results for markers in NZ/Aus isolates is biologically relevant, then clonality still exists but is limited to isolates from North America, US/MX/CR.
Here is one way to look at the data. In Table S1, first exclude results from poorly performing marker Ut497. Then, check if results from at least five out of the remaining seven markers are consistent, we obtain 9 US, 5 CR and 57 MX isolates, but only 2 NZ (NZ99990, NZ71696) and 4 Aus (VPRI 21344, VPRI 21238, VPRI 20858, VPRI 20881) isolates. Based on Figure 1, introductions in Aus/NZ preceded US/MX/CR. Could these isolates in US/MX/CR be derived from Aus/NZ?

If so, then the title could be modified to "Invasions of gladiolus rust in North America are caused by a widely-distributed clone of Uromyces transversalis"

4. Although Ut497 was a poor performing marker, it surprised me that it did not work on three US isolates, including two of the sequenced isolates.

5. Explain how scoring is done for the multiplex reactions either in Results or Methods section, and what does 'a small peak below the threshold peak size of 500' mean.

6. What is the implication of a NOPEAK? What does NOPEAK mean? Does it mean that there was a PCR product visible on the gel but no detection during fragment analysis, or, no product could be visualized on the gel.

7. Table 4: Why these six markers? This analysis can be done for all the 25, even the original 60, markers, as the genomes sequences are available for the three isolates.

Minor concerns
----- --------
1. In the whole genome sequencing, both the dikaryons have been sequenced. Reads can be mapped back to the assembly to find SNPs in regions of interest.

2. It would help to list a few examples where allelic and genotypic diversity based on SSRs was in agreement with results obtained from other types of markers such as or SNPs derived from GBS or whole genomes.

3. Lines 259-260: Was the number of shared contigs higher in the two FL11 isolates versus the CA11?

4. Line 279: What criteria was used to select these eight markers? It is explained in the Methods but not in results. Also please expand on the multiplexing scheme in the results.

5. Being a dikaryon, is it a surprise that most markers were polymorphic?

6. Line 349: What is the basis of this calculation? Please elaborate.

6. Table 1: Make the state/Country as a separate column

7. Table 2: Include the number of markers in the heading

8. Table 3: Please add another column that lists the total reads per isolate. Then change the column headings to reflect Filtered for quality, and Reads that assembled.

9. Table S1: It should be cited in the 'Genotypic analyses' section in Results.
Add a footnote on what these numbers means. Also, add a footnote on how is 'NA' different from 'NOPEAKS'.

267: Where is the term 'electropherogram' defined?
283: Isolate(s) from South Africa not listed in Table 1
289: Where has been the 'scoring threshold peak size 500' been defined?
296: What was the rationale for choosing a monomorphic marker?

TYPOs
242: Remove the double space - " reads " to " reads "
243: 'insert size', not 'insertion size'
244: Expand PF
245: Either list three numbers or remove 'respectively'
248: Add percentages along with the number of reads that were assembled
249: Remove '(contigs)'
259: Delete 'microsatellite containing'
264: Add 'sequenced', as in three sequenced isolates

·

Basic reporting

DeLong et al. provide a clear manuscript about the efforts that led to the identification of high degrees of clonailty on an important Urediniomycete affecting Gladiolus species.

The document is clear, professional and very well written. Each of the sections is well documented and include all necesary information for a good report. The authors do a great job at providing sufficient background information to guide the reader throughout the introduction while also stating the research objective clearly.

While the evidence presented is clear and objective, I feel that the only shortcoming is the absence of a main figure that can summarize the results in a simple and clear manner. In my opinion, a revised version of Supplementary Figure 1 would suffice to present a summary of the results and the main findings.

Experimental design

The experimental design presented by DeLong et al. is solid and adequate for the research question posed. The knowledge gap is clearly stated and the methods used in the current manuscript aim to fill this gap.

Minor comments on stating parameters and tools for repeatability have been added to a revised version to the manuscript, but none of these comments convey a major concern for the experimental procedures used in this research.

Validity of the findings

The findings of this manuscript present a puzzling but common feature of fungal plant pathogenic organisms: A link between clonal reproduction and large scale movement of these disease causing organisms.

The results found here are very relevant to the study of the evolution of fungal organisms, and provide more evidence on the uniqueness of the evolutionary patterns of fungal and fungal-like organisms. All the results are well supported and solve the proposed research questions by the authors.

While the current era of genomics seems to be moving fast into whole genome data genotyping techniques, reports like this are important as they remind researchers that traditional markers are still useful and necessary in plant pathology or other sciences that require markers that can be screened rapidly and are cost efficient.

There are some minor comments attached on the presentation of some of the discussion/conclusions, mostly on the extrapolation of the results from microsatellite data with the biology and pathogencity of the organism. These comments should be addressed so the authors can also provide a critical view of their findings as well as future perspectives of what can be done to evaluate if the lack of genetic diversity found in microsatellite data truly reveals a general lack of divergence in this species, or if other techniques can provide a more accurate view of the evolution of these organisms.

These comments are minor and don't compromise the structure of the manuscript nor the main findings, but are necessary for a more objective view of some of the findings presented here.

Additional comments

Overall, the manuscript presents a very interesting finding on this fungal species. The manuscript is very well written and easy to understand; the methods are solid and the results support the hypothesis proposed.

A few minor comments have been added to the manuscript. Mainly, the absence of a results-driven figure makes gives the manuscript a feeling of incompleteness. However, Sup. Fig. 1 can be slightly improved and presented as evidence for the main findings of this report. Subsequently, some of the speculation of the lack of polymorphisms in SSR v.s. clonality of the species need to be either better justified or be presented with more carefulness. Other minor comments are added to the manuscript directly.

None of these minor comments affect the overall findings. Great manuscript overall.

---

## Round 0.2 · accepted · Accept

Thank you for your revisions which I feel have addressed the reviewers concerns.